# Targeting NKG2DL with Bispecific NKG2D–CD16 and NKG2D–CD3 Fusion Proteins on Triple–Negative Breast Cancer

**DOI:** 10.3390/ijms241713156

**Published:** 2023-08-24

**Authors:** Polina Kaidun, Samuel J. Holzmayer, Sarah M. Greiner, Anna Seller, Christian M. Tegeler, Ilona Hagelstein, Jonas Mauermann, Tobias Engler, André Koch, Andreas D. Hartkopf, Helmut R. Salih, Melanie Märklin

**Affiliations:** 1Clinical Collaboration Unit Translational Immunology, German Cancer Consortium (DKTK), Department of Internal Medicine, University Hospital Tuebingen, 72076 Tuebingen, Germany; polina.kaidun@med.uni-tuebingen.de (P.K.); samuel.holzmayer@med.uni-tuebingen.de (S.J.H.); sarah.maria.greiner@med.uni-tuebingen.de (S.M.G.); anna.seller@med.uni-tuebingen.de (A.S.); ilona.hagelstein@med.uni-tuebingen.de (I.H.); jonas.mauermann@med.uni-tuebingen.de (J.M.); helmut.salih@med.uni-tuebingen.de (H.R.S.); 2Cluster of Excellence iFIT (EXC 2180) ‘Image–Guided and Functionally Instructed Tumor Therapies’, Eberhard Karls University Tuebingen, 72074 Tuebingen, Germany; 3Department of Obstetrics and Gynecology, University Hospital Tuebingen, 72076 Tuebingen, Germany; christian.tegeler@med.uni-tuebingen.de (C.M.T.); tobias.engler@med.uni-tuebingen.de (T.E.); andre.koch@med.uni-tuebingen.de (A.K.); andreas.hartkopf@med.uni-tuebingen.de (A.D.H.)

**Keywords:** NKG2D, bispecific, fusion protein, TNBC, breast cancer, NK cell, T cell, immunotherapy

## Abstract

Triple–negative breast cancer (TNBC) is a particularly aggressive subtype of breast cancer with a poor response rate to conventional systemic treatment and high relapse rates. Members of the natural killer group 2D ligand (NKG2DL) family are expressed on cancer cells but are typically absent from healthy tissues; thus, they are promising tumor antigens for novel immunotherapeutic approaches. We developed bispecific fusion proteins (BFPs) consisting of the NKG2D receptor domain targeting multiple NKG2DLs, fused to either anti–CD3 (NKG2D–CD3) or anti–CD16 (NKG2D–CD16) Fab fragments. First, we characterized the expression of the NKG2DLs (MICA, MICB, ULBP1–4) on TNBC cell lines and observed the highest surface expression for MICA and ULBP2. Targeting TNBC cells with NKG2D–CD3/CD16 efficiently activated both NK and T cells, leading to their degranulation and cytokine release and lysis of TNBC cells. Furthermore, PBMCs from TNBC patients currently undergoing chemotherapy showed significantly higher NK and T cell activation and tumor cell lysis when stimulated with NKG2D–CD3/CD16. In conclusions, BFPs activate and direct the NK and T cells of healthy and TNBC patients against TNBC cells, leading to efficient eradication of tumor cells. Therefore, NKG2D–based NK and T cell engagers could be a valuable addition to the treatment options for TNBC patients.

## 1. Introduction

Invasive breast cancer (IBC) is the most common malignancy in women worldwide [1]. The absence of estrogen receptor, progesterone receptor and human epidermal growth receptor 2 (HER2) is classified as triple–negative breast cancer (TNBC). TNBC accounts for up to 20% of all breast cancers, and remains one of the most difficult to cure. It also exhibits a particularly aggressive phenotype compared to other subtypes of IBC [2,3]. TNBC patients typically receive conventional chemotherapy [4,5], but the recent introduction of PD1/PDL1 checkpoint inhibitors such as Pembrolizumab and Atezolizumab enhance T–cell–mediated anti–tumor effects and improve outcomes in TNBC. However, recurrence rates remain high, and the management of metastatic disease remains challenging [6]. 

The natural killer group 2D ligand (NKG2DL) family consists of the MHC class I–related chain (MIC) MICA and MICB and the UL16–binding proteins (ULBP) ULBP1–6. NKG2DLs are generally absent on healthy tissues, but are induced upon cellular stress such as malignant transformation or infection [7]. NKG2DLs are potent inducers of antitumor immunity by activating natural killer (NK) cells and T cells after binding to the NKG2D receptor. This leads to the release of effector granules such as perforin and granzyme, thereby inducing apoptosis of the tumor cells [8]. NKG2DL expression can be further enhanced by treatment with various DNA–damaging chemotherapeutic agents such as doxorubicin [9]. It has been reported that all TNBCs express NKG2DL [10]. Therefore, NKG2DLs are promising tumor antigens, and there are many therapeutic approaches targeting the NKG2DL/NKG2D axis are currently under clinical investigation. These include chimeric antigen receptor T cells (CAR–T) or chimeric antigen receptor NK cells (CAR–NK), or an anti–MICA/MICB monoclonal antibody, which have induced partial responses in leukemia and multiple myeloma patients (NCT04550663, NCT05117476, NCT05117476) [11,12]. We have previously targeted NKG2DLs with NKG2D fusion proteins, in which the extracellular domain of the NKG2D receptor was fused to an antibody–dependent cellular cytotoxicity (ADCC)–optimized Fc–IgG1 moiety. With this construct, we were able to successfully induce NK cell reactivity and induce lysis of IBC cells [13]. In general, monoclonal antibodies (mAbs) and IgG fusion proteins face two problems. First, they can only activate NK cells as effector cells, and second, their Fc parts can also bind to inhibitory Fcγ receptors, which reduces their cytotoxic effect [14]. To overcome these hurdles, we improved the anti–tumor activity by creating bispecific fusion proteins (BFPs) with the NKG2D receptor domain linked to anti–CD16 or anti–CD3 Fab fragments to direct the NK cells or cytotoxic T lymphocytes against tumor cells. The main advantage of these two constructs is, on the one hand, that binding to inhibitory Fcγ receptors is prevented by the anti–CD16 part and, on the other hand, that T cells with higher cytotoxic potential, can be recruited by the anti–CD3 part. These BFPs have already been shown to have potent anti–tumor effects against soft tissue sarcomas and acute myeloid leukemia [15,16]. 

In this study, we evaluate the efficacy of NKG2D–CD16 and NKG2D–CD3 bispecific fusion proteins (BFPs) in the treatment of TNBC. We show that the BFPs effectively direct NK cells and T cells to the tumor cells and eliminate the TNBC cells. Furthermore, we report that NK cells and T cells from TNBC patients currently undergoing chemotherapy are able to efficiently eliminate TNBC tumor cells.

## 2. Results

### 2.1. Characterization of NKG2DL Expression in Triple–Negative Breast Cancer Cells 

We examined the mRNA expression of NKG2DL in eight TNBC cell lines and found that all of them expressed at least two different ligands at varying levels (Figure 1A). *MICA* showed the highest expression, while the expression of *MICB* and *ULBP1–3* varied between different cell lines. *ULBP4* was mostly absent in all cell lines tested. 

Next, we determined the surface protein expression using specific mAbs against MICA, MICB, and ULBP1–4 (Figure 1B). MICA and ULBP2 showed the highest surface expression among cell lines tested, whereas ULBP4 could not be detected (Figure 1B). Six TNBC cell lines expressed four or five different NKG2DLs (37.5% each), except HS–578T, which expressed only two, and BT–549, which expressed three types of NKG2DLs (12.5% each) (Figure 1C and Appendix A). We examined the binding of all NKG2DLs on the surface with the extracellular NKG2D receptor domain fused to the Fc–IgG1 (Figure 1D). Based on our results, we selected BT–549, with the lowest, MDA–MB–468, with intermediate, and MDA–MB–231, with high NKG2DL expression for further experiments (Figure 1E and Appendix A). To evaluate the expression of NKG2DL in primary breast cancer material (n = 200), we used the TCGA database analysis and observed predominant expression of *MICA*, *MICB* and *ULBP2* (Figure 1F).

### 2.2. Effector Cell Recruitment by NKG2D BFPs against TNBC Cells

We analyzed the ability of NKG2D–CD16 and NKG2D–CD3 to recruit NK cells and T cells, respectively, to the TNBC cells. MDA–MB–231 and MDA–MB–468 cells were incubated with peripheral blood mononuclear cells (PBMCs) from healthy donors in the presence or absence of NKG2D–CD16. Immune cell recruitment to TNBC was quantified visually by determining the number of colocalized effector and target cells per field of view (FoV), a representative example of which is shown for MDA–MB–231 (Figure 2A,B). We observed a significant increase in CD16^+^ cells interacting with TNBC cells when treated with NKG2D–CD16 (Figure 2C and Appendix A). A similar effect was observed for CD3^+^ cells when MDA–MB–231 or MDA–MB–468 cells were co–incubated with PBMCs from healthy donors in the presence of NKG2D–CD3 (Figure 2D–F and Appendix A), confirming the effective recruitment of NK cells and T cells induced by the respective BFPs.

### 2.3. Modulation of NK Cell and T Cell Reactivity against TNBC Cells with PBMCs from Healthy Donors

To analyze whether our constructs were able to induce effector cell reactivity against TNBC cells, we cocultured the PBMCs from healthy donors with BT–549, MDA–MB–231 and MDA–MB–468 cells in the presence or absence of BFPs. Application of NKG2D–CD16 or NKG2D–CD3 resulted in a significant induction of the activation marker CD69 on NK cells and T cells (Figure 2G–J). Similarly, the increased surface expression of CD107a confirmed that NKG2D–CD16/CD3 strongly induced the degranulation of NK cells and T cells, respectively (Figure 3A–D). Analysis of the supernatants by Legendplex assays showed a significant increase in IFNγ, TNF, granzyme A, perforin and granulysin secretion after treatment with NKG2D–CD16 (Figure 3E) or NKG2D–CD3 (Figure 3F). 

### 2.4. Induction of Target Cell Lysis by Bispecific NKG2D Fusion Proteins

We then analyzed whether activation of effector cells resulted in cytotoxicity against tumor cells. Naturally, NK cells and T cells have different kinetics of activation and tumor cell killing [17]. To investigate the intrinsic function of both types of effector cells, we performed cell lysis assays with different incubation times. Europium–based short–term cytotoxicity assays revealed the potency of both BFPs to lyse target cells after 2 h, but the more pronounced effect was observed with NKG2D–CD16 (Figure 3G,H and Appendix A). At 72 h, flow cytometry–based lysis assays showed much stronger lysis with NKG2D–CD3 (Figure 3I,J and Appendix A). The finding that the lysis ability of NKG2D–CD3 emerged after prolonged incubation time and then surpassed the effects of NKG2D–CD16 was also confirmed by live cell imaging over an incubation period of 120 h (Figure 3K,L and Appendix A). 

### 2.5. Reactivity of NK Cells and T Cells from TNBC Patients against TNBC Cells 

To understand whether there are disease– and treatment–related changes in the immune cell subsets of TNBC patients that could affect the treatment efficacy of our BFPs, we compared the PBMCs from healthy donors and TNBC patients. At the time of PBMC collection, all tested TNBC patients studied were receiving cytotoxic chemotherapy with or without added immunotherapy. The specific details of the clinical characterization and treatment regimens of all TNBC patients are shown in Table 1. In our TNBC cohort, we observed normal counts of lymphocytes, leukocytes, neutrophils, monocytes and platelets (Figure 4A and Appendix A). We evaluated the distribution of the PBMC subsets, including T cells (CD3^+^CD4^+^ and CD3^+^CD8^+^), monocytes (CD14^+^), dendritic cells (DCs, CD3^−^CD56^−^CD14^−^CD19^−^HLA–DR^+^), B cells (CD19^+^), NKT (CD3^+^CD56^+^) and NK cells (CD3^−^CD56^+^), by performing flow cytometric analysis. TNBC samples showed fewer CD4^+^ T cells but the same amount of CD8^+^ T cells and an increase in DCs. No relevant difference was observed for other mononuclear cells compared to healthy donors (Figure 4B). To determine whether lymphocytes from TNBC patients could be stimulated by NKG2D–CD16 and NKG2D–CD3 in a similar manner compared to healthy donors, we used cocultures with TNBC cells. Treatment with both NKG2D–CD16 and NKG2D–CD3 significantly increased the activation of NK cells and CD4^+^ and CD8^+^ T cells, as determined by flow cytometry, for CD69 surface expression at 24 h (Figure 4C and Appendix A). In line, the detection of CD107a upregulation confirmed the effective degranulation of NK cells and T cells after treatment with BFPs (Figure 4D and Appendix A). Analysis of the coculture supernatants revealed a significant increase in the release of IFNγ, TNF, granzyme A, perforin and granulysin after treatment with both NKG2D–CD16 (Figure 4E) and NKG2D–CD3 (Figure 4F). 

Finally, we evaluated the killing capacity of NK cells and T cells from TNBC patients against TNBC cell lines by live cell imaging (Figure 4G). While the strong lysis effect of NKG2D–CD3 was still present in TNBC patients, NKG2D–CD16 was only sufficient for the MDA–MB–468 target cell line (Figure 4G). 

## 3. Discussion

The landscape of cancer treatment options has evolved rapidly in recent years. With the success of checkpoint inhibitors in TNBC, immunotherapy has become a standard of care, highlighting the strength of T–cell–based strategies for anti–tumor therapy. Despite these promising developments, survival rates are still lower than in other breast cancer subtypes, and the risk of recurrence remains a challenge [2]. NKG2D–based NK cell and T cell engagers could be potential options for the treatment of TNBC.

In our present study, we observed effective lysis of TNBC cells by PBMCs from both healthy donors and TNBC patients after treatment with NKG2D–CD16 and NKG2D–CD3 BFPs. Increased surface expression of CD69 and CD107a on NK cells and CD4^+^ and CD8^+^ T cells indicated enhanced activation and degranulation accompanied by cytokine release upon NKG2D–CD3/CD16 BFP treatment. This effect was reflected in significant cytotoxicity against TNBC cells. 

Recently, maturation of NK cells has been shown to be exclusively important in predicting their potential effect against TNBC tumors [18]. In humans, terminally differentiated peripheral blood NK cells, exhibiting the CD56^dim^CD16^+^ phenotype, are cytotoxic. In contrast, CD56^bright^CD16^−^ NK cells, found in secondary lymphoid tissues, are considered to be immature and have reduced cytotoxic potential [19]. Thacker et al. demonstrated an increased number of immature NK cells in various TNBC models and provided evidence for the pro–tumorigenic nature of NK cells within the tumor microenvironment (TME) of TNBC [18]. Using the NKG2D–CD16 construct, we recruited cytotoxic CD16^+^ NK cells, potentially altering the balance between immature and cytotoxic NK cell populations. Another advantage of NKG2D–CD16 is that NK cell activation is not dependent on the NKG2D receptor, which is beneficial given that NKG2D expression on NK cells can be downregulated under certain circumstances [20]. It is noteworthy that NKG2D–CD16 demonstrated superior cytotoxicity in the short–term assays, whereas NKG2D–CD3 required a longer period to achieve maximal anti–tumor activity. Although both NKG2D–CD16 and NKG2D–CD3 were effective, the killing ability of T cells stimulated by NKG2D–CD3 was more pronounced. This difference could be attributed to the fact that NKG2D–CD3 can induce T cell proliferation, whereas no proliferation was observed in NK cells activated by NKG2D–CD16 [21]. 

In terms of clinical implications, it is plausible to assume that the lower cytokine release upon treatment with NKG2D–CD16 compared to NKG2D–CD3 may lead to fewer side effects. This aspect qualifies NKG2D–CD16 as a promising therapy for elderly TNBC patients who may not tolerate immediate massive immune activation. In addition, our study showed that PBMCs from TNBC patients undergoing chemotherapy showed notable activation of NK cells and T cells upon BFP stimulation, suggesting the feasibility of combined immunotherapy. However, additional data are needed to fully understand the bioavailability of NKG2D BFPs in the TNBC TME in the clinical setting. 

In conclusion, these results highlight the encouraging antitumor efficacy of NKG2D–CD3 and NKG2D–CD16 BFPs. PBMCs from TNBC patients showed a robust potential to targeting cancer cells, even during ongoing chemotherapy, suggesting that our novel approach could serve as stand–alone therapy or as an adjunct to chemotherapy.

## 4. Materials and Methods

### 4.1. Cell Lines

The human cell lines CAL–51, HS–578T, HCC70, HCC1500, BT–549, MDA–MB–157, MDA–MB–231, and MDA–MB–468 were obtained from DSMZ or ATCC. Green fluorescent protein (GFP)–expressing breast cancer cell lines (BT–549, MDA–MB–231 and MDA–MB–468) were established by infection with Incucyte nuclight green lentivirus reagent (EF1α, puro) (Sartorius Group, Göttingen, Germany) and sorted by flow cytometry. All cell lines were routinely tested for mycoplasma contamination. 

### 4.2. qRT–PCR 

RNA from 1 million TNBC cells was isolated using the High Pure RNA Isolation Kit (Roche, Basel, Switzerland) and tested for quality and quantity using the NanoDropTM One/OneC (Thermo Fisher, Carlsbad, CA, USA). cDNA synthesis was performed using a FastGeneScriptase II reverse–transcription PCR kit (Nippon Genetics Europe, Dueren, Germany), and 10 ng of cDNA was used for quantitative PCR. Previously described primers for *MICA* (FWD: 5′-ggcgcctaaagtctgagaga-’3, REV: 5′-aaccctgactgcacagatcc-′3), *MICB* (FWD: 5′-ctgagaaggtggcgacgta-′3, REV: 5′-cgaagactgtggggctca-′3), *ULBP1* (FWD: 5′-actgggaacaaatgctggat-′3, REV: 5′-gagaaggctccagggactg-′3), *ULBP2* (FWD: 5′-ccgctaccaagatccttctg-′3, REV: 5′-gggatgacggtgatgtcatag-′3), *ULBP3* (FWD: 5′-tccctggcatctgagaagag-′3, REV: 5′-cagaaaggcacagtggtgagt-′3), *ULBP4* (FWD: 5′-agcacttggggagaattgac-′3, REV: 5′-cttgcagagtggaaggatcac-′3) and *GAPDH* (FWD: 5′-agccacatcgctcagacac-′3, REV: 5′-gcccaatacgaccaaatcc-′3) were used for quantitative PCR and were performed using Perfecta SYBR Green FastMix (Quanta Biosciences Beverly, MA, USA) measured on a LightCycler480 (Roche, Basel, Switzerland) [13,21].

### 4.3. Flow Cytometry

Single NKG2DL–specific mAbs (MICA, clone: AMO1; MICB, clone: BMO1; ULBP1 clone: AUMO3; ULBP2, clone: BUMO1; ULBP3, clone CUMO3; ULBP4, clone: DUMO) or corresponding isotype controls (10 µg/mL) followed by goat anti–mouse PE conjugate (Dako, Glostrup, Denmark) were used for staining, as described previously [13]. To study the binding of NKG2Ds, we used biotinylated NKG2D–Fc proteins and an isotype control (10 µg/mL) (both R&D systems, Minneapolis, MN, USA), stained with streptavidin–PE conjugate (LifeTechnologies, Carlsbad, CA, USA), as previously described [21]. 

PBMC subsets from TNBC patients and healthy donors were identified by counterstaining with CD3–APC/Fire750 (clone: SK7), CD4–Pacific Blue (clone: RPA–T4), CD8–BV605 (clone: RPA–T8), CD14–BV785 (clone: M5E2), CD16–APC (clone: 3G8), CD19–FITC (clone: HIB19), CD56–PeCy7 (clone: HCD56) and HLA–DR–BV650 (clone: L243) (all BioLegend, San Diego, CA, USA) and 7–AAD (Biolegend, San Diego, CA, USA)). 

NK cell and T cell activation and degranulation were assessed by flow cytometry. Briefly, 100,000 TNBC cells were cultured with allogeneic PBMCs from healthy donors or TNBC donors at an effector to target (E:T) ratio of 2.5:1 for 4 h and 24 h, followed by mAb staining. For flow cytometric evaluation of specific target cell lysis, TNBC cells were stained with 2.5 µM CellTrace™ Violet (Thermo Fisher Scientific, Waltham, MA, USA) and cocultured with PBMCs (E:T ratio of 5:1) in the presence of the BFPs (2.5 µg/mL) or control for 72 h. Latex beads (Sigma–Aldrich, Darmstadt, Germany) were used to ensure that equal volumes of cell suspension were analyzed. 7–AAD was used to exclude dead cells from the analysis. PBMC subsets from healthy donors and TNBC patients were identified by counterstaining. 

All measurements were performed using a FACS Canto II, FACS Fortessa or FACS Aria III (BD Biosciences, Heidelberg, Germany) and data were analyzed using FlowJo–V10 software (BD Biosciences, Heidelberg, Germany). 

### 4.4. Production and Purification of Bispecific NKG2D Fusion Proteins

To generate NKG2D–CD3 and NKG2D–CD16 BFPs, the extracellular domain of NKG2D (F78–V216) was fused C–terminally to the heavy chain of a Fab fragment specific for either CD3 (clone UCHT1) or CD16 (clone 3G8) using a CH2 linker [22,23]. The CH_2_ domain of IgG1 was modified to attenuate FcγR and glycan receptor binding, complement fixation and reduce immunogenicity. The following amino acids in the CH_2_ domain of IgG1 have been swapped or deleted: E233→P; L234→V; L235→A; G236→deleted; D265→G; N297→Q; A327→Q; A330→S [24,25]. The N297→Q modification prevents the addition of a glycan structure and C226 and C229 were replaced with serine to prevent dimerization [25,26]. Final proteins were produced after transfection of SP2/0–Ag14 cells (American Type Culture Collection, Manassas). Subcloned transfectants were cultured in IMDM (GIBCO, Carlsbad, CA, USA) supplemented with 10% of fetal bovine serum (PAN–biotech, Aidenbach, Germany), 1% non–essential amino acid solution (Sigma–Aldrich, St. Louis, MO, USA), 1% L–glutamine (PAN–biotech, Aidenbach, Germany), 1% sodium pyruvate solution (Sigma–Aldrich, St. Louis, MO, USA), 1% penicillin–streptomycin (PAN–biotech, Aidenbach, Germany), 1 mg/mL G418 (Invitrogen, Carlsbad, CA, USA). BFPs were purified from culture supernatants by HiTrap KappaSelect™ affinity chromatography (GE Healthcare, Chicago, IL, USA) followed by preparative size exclusion chromatography on Superdex HiLoad 16/60 column (GE Healthcare, Chicago, IL, USA). Purity was determined by 4–12% non–reducing gradient SDS–PAGE (Invitrogen, Carlsbad, CA, USA). Endotoxin (EU) levels were tested using Endonext^TM^ (Biomerieux, Nuertingen, Germany) and were ≤1 EU/mg for all proteins (Appendix A).

### 4.5. Immunofluorescence

For immunofluorescence staining, MDA–MB–231 and MDA–MB–468 cells were incubated with PBMCs from healthy donors at an E:T ratio of 2.5:1. After 30 min of incubation, the media was removed and the cells were fixed with 4% paraformaldehyde. Cells were washed with PBS and blocked with 5% bovine serum albumin blocking solution containing 0.1% Tween20 and 0.2% Triton X–100 for 60 min. Staining was performed with murine mAb αCD3 (clone OKT3, 1:25 Biolegend, San Diego, CA, USA), αCD16 (clone #1001049, 1:25, R&D Systems, Minneapolis, MN, USA) and rabbit mAb α–tubulin (clone 11H10 1:500, Cell Signaling, Danvers, MA, USA). Primary antibodies were detected with Alexa–Fluor488–conjugated anti–mouse (1:500) and Alexa–Fluor594–conjugated anti–rabbit (1:500) antibodies (both Invitrogen, Carlsbad, CA, USA). DAPI was used for nuclear staining. Images were captured with an LSM800 microscope (Zeiss, Oberkochen, Germany) and analyzed with ImageJ 1.53a software. 

### 4.6. Analysis of Cytokine Secretion

To evaluate the cytokine release, supernatants from the coculture assays were analyzed at 4 h and 24 h using the Legendplex Human CD8/NK Panel (BioLegend, San Diego, CA, USA) according to the manufacturer’s protocol.

### 4.7. Cytotoxicity Assay

Lysis of TNBC cells by PBMCs from healthy donors in the presence or absence of the BFP (2.5 µg/mL) was assessed by a 2 h Europium–based cytotoxicity assay, as previously described [15]. Briefly, TNBC cells were labeled with DELFIA^®^ BATDA (Perkin Elmer, Waltham, MA, USA) for 30 min. After labeling, the cells were incubated with PBMCs at the indicated E:T ratios. After 2 h of incubation, 20 µL of supernatant from each sample was mixed with 200 µL DELFIA^®^ Europium solution (PerkinElmer, Waltham, MA, USA). Subsequent samples were measured using a Spectra Max ID5 system (Molecular Devices, Silicon Valley, CA, USA). Specific lysis was calculated as follows:100 × (experimental release − spontaneous release)/(maximum release − spontaneous release)

Long–term cytotoxicity analyses were performed using the IncuCyte^®^ S3 Live–Cell Analysis System (Essenbioscience, Sartorius, Göttingen, Germany). GFP–expressing TNBC cells were cultured with PBMCs from healthy donors (E:T ratio of 5:1) with or without the indicated treatments (2.5 µg/mL each). Live cell images were taken every 3–4 h at 10x magnification. To quantify live cells, the total green area of each variant was normalized to the corresponding measurement at T = 0 h.

### 4.8. Primary Material

PBMCs from healthy donors and TNBC patients were isolated by density gradient centrifugation. Blood samples from 19 consecutive TNBC patients treated at the Department of Gynecology at the University Hospital of Tübingen were included in our study. All samples were collected in January/February 2023.

### 4.9. Statistics

Unless otherwise noted, values are mean ± standard error of the mean (SEM). For continuous variables, in the case of normal distribution, Student’s *t*–test was used for normal distribution and Mann–Whitney U test for non–normal distribution. GraphPad Prism 9.4.1 was used for statistical analysis. All statistical tests were considered significant if the *p*-value was less than 0.05 (* *p* < 0.05, ** *p* < 0.01, *** *p* < 0.001).

## Figures and Tables

**Figure 1 ijms-24-13156-f001:**
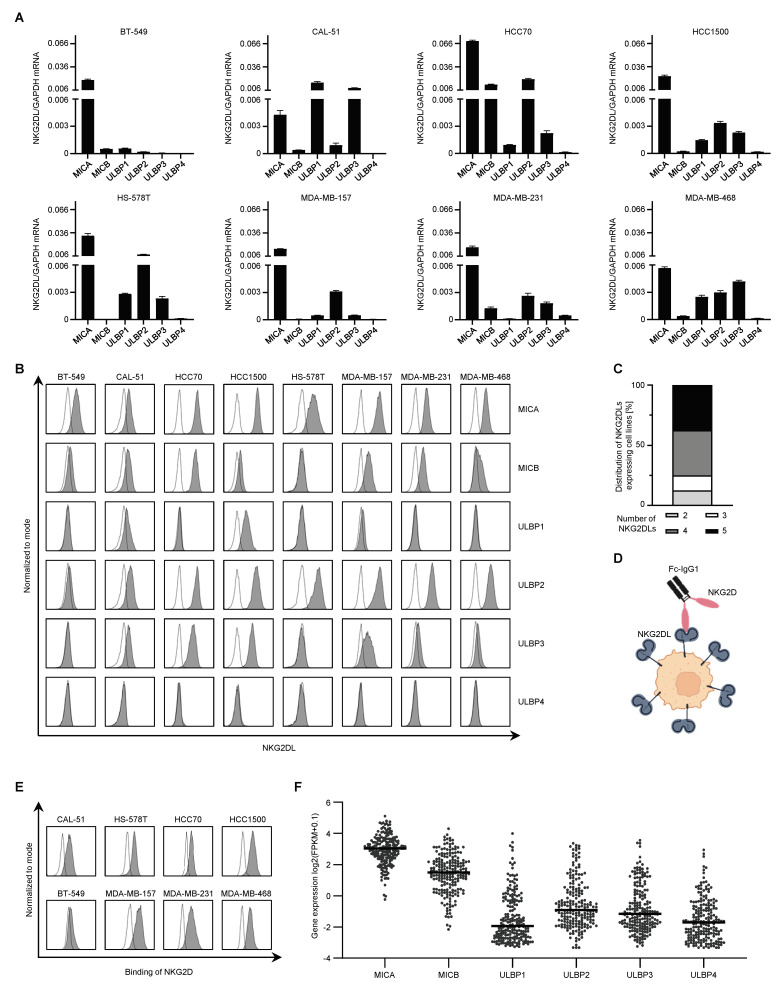
**Expression of NKG2DL in triple–negative breast cancer.** (**A**) Expression of various NKG2DL mRNAs in TNBC cell lines was measured by qRT–PCR. (**B**) Surface expression of the indicated NKG2DL was stained with specific mAbs (10 µg/mL) and assessed by flow cytometry. Open histograms show isotype control staining, and filled histograms show the staining of the indicated NKG2DL. (**C**) The relative proportion of cell lines expressing the indicated amount of different NKG2DL on the surface is shown. (**D**) Schematic representation of NKG2DL staining with the NKG2D–Fc in which the disulfide–linked homodimer of the extracellular NKG2D domain is fused to the Fc portion of an IgG1 antibody (created with BioRender.com). (**E**) Binding of NKG2D to the surface of the TNBC cell lines was assessed by staining with an Fc fusion protein (10 µg/mL). Open histograms show isotype control staining, and filled histograms show the staining of the indicated NKG2DL. (**F**) Relative expression of MICA, MICB and ULBP1–4 RNA in breast cancer patients (n = 200), assessed using TCGA datasets (FPKM: fragments per kilobase of transcripts per million mapped fragments).

**Figure 2 ijms-24-13156-f002:**
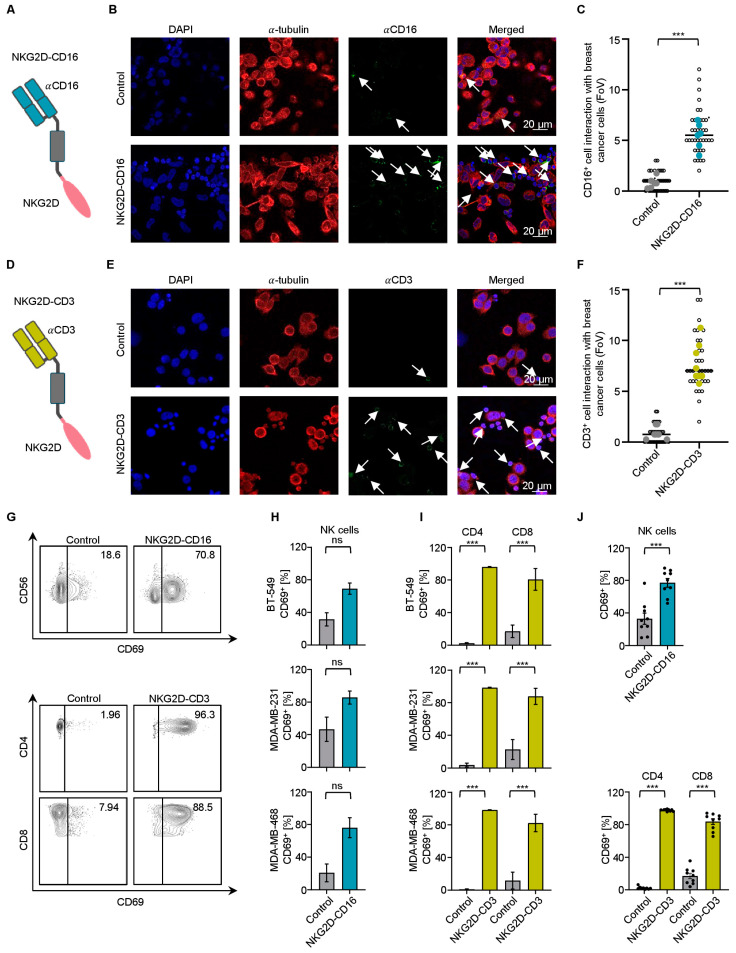
**Recruitment of effector cells to the TNBC cell lines and activation of NK cells and T cells by BFPs.** (**A**,**D**) Schematic illustration of BFPs with the αCD16 (blue) (**A**) or αCD3 (green) (**D**) sFab linked with a CH2 domain of IgG1 to the extracellular receptor domain of NKG2D (created with BioRender.com). (**B**,**C**,**E**,**F**) TNBC tumor cells were incubated with PBMCs from a healthy donor (E:T ratio of 5:1) and control or NKG2D–CD16/CD3 (2.5 µg/mL) for 30 min. (**B**,**E**) Representative images for (**B**) MDA–MB–231 cells treated with control or NKG2D–CD16 with CD16^+^ cells in green and (**E**) MDA–MB–468 cells treated with control or NKG2D–CD3 with CD3^+^ cells in green. α–Tubulin is shown in red and DAPI (300 nM) in blue was used to counterstain the nucleus. White arrows indicate the NK cells (**B**) and T cells (**E**) engaged with targets. (**C**,**F**) Pooled data of PBMCs from healthy donors (n = 4) with n = 4 fields of view (FoVs) per donor (small dots for each FoV, large dots indicate the mean per donor) incubated with NKG2D–CD16 (blue) and NKG2D–CD3 (green) are shown. (**G**–**J**) Activation of NK cells (CD56^+^) and T cells (CD4^+^ and CD8^+^) was determined by the expression level of CD69 after 24 h of PBMCs from healthy donors in coculture with TNBC cells at an E:T ratio of 2.5:1 in the presence or absence of NKG2D–CD16 (blue) or NKG2D–CD3 (green) (both 2.5 µg/mL). (**G**) Exemplary flow cytometry results for CD69 obtained with BT–549 cells are shown. (**H**) NK cell or (**I**) CD4^+^ and CD8^+^ T cell activation with PBMCs from healthy donors (n = 3) with the indicated TNBC cell lines are shown. (**J**) Combined data of the TNBC cell lines BT–549, MDA–MB–231 and MDA–MB–468 cocultured with PBMCs from healthy donors (n = 3). All statistical tests were considered significant if the *p*-value was below 0.05 (*** *p* < 0.001). *p*-values above 0.05 are marked as not significant (ns).

**Figure 3 ijms-24-13156-f003:**
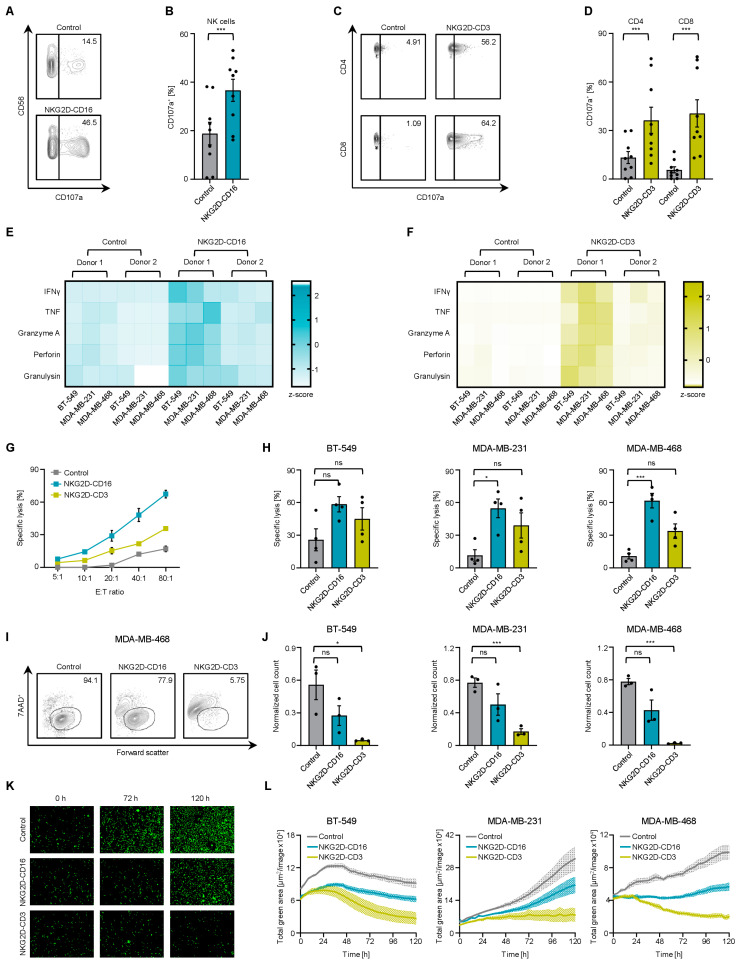
**Reactivity of healthy donor T and NK cells against TNBC cell lines in response to NKG2D–CD16/CD3 treatment.** PBMCs from healthy donors were cultured with TNBC cells at an E:T ratio of 2.5:1 (unless otherwise noted) with controls (gray) or NKG2D–CD16 (blue) NKG2D–CD3 (green) (both 2.5 µg/mL). (**A**–**D**) Degranulation of NK cells after the treatment with NKG2D–CD16 and T cells after the treatment with NKG2D–CD3 was determined by the expression level of CD107a after 4 h. (**A**,**C**) Exemplary flow cytometry results for (**A**) NK cells and (**C**) T cells CD107a^+^ obtained with MDA–MB–468 cells are shown. (**B**,**D**) Combined data with the TNBC cell lines BT–549, MDA–MB–231 and MDA–MB–468 and with PBMCs from healthy donors (n = 3) are shown. (**E**,**F**) Cytokines and effector molecules released into the supernatant after treatment with (**E**) NKG2D–CD16 after 4 h and (**F**) NKG2D–CD3 after 24 h coculture of TNBC cell lines (n = 3) with PBMCs from healthy donors (n = 2) measured by Legendplex assay. (**G**–**L**) Cytotoxicity of NKG2D–CD16/CD3 against TNBC cell lines was evaluated. (**G**,**H**) Cell lysis determined by 2 h Europium killing assay is shown. (**G**) Exemplary lysis of MDA–MB–468 with a healthy PBMCs donor at the indicated E:T ratios and (**H**) cytotoxicity against the indicated TNBC cell lines by PBMCs from healthy donors (n = 3) at an E:T ratio of 80:1 are shown. (**I**) Exemplary FACS data of MDA–MB–468 with a healthy donor PBMC and (**J**) lysis of the indicated TNBC cell lines determined by flow cytometry–based lysis assay (E:T ratio of 5:1) are shown. (**K**,**L**) Cell death of TNBC cells as determined by a live cell imaging system. Green fluorescent target cells were incubated with BFPs and PBMCs from healthy donors (n = 3) at an E:T ratio of 5:1 for 120 h. (**K**) Representative images at 0 h, 72 h and 120 h at 10x magnification and (**L**) data for separate TNBC cell lines and PBMCs from healthy donors (n = 3) are shown. All statistical tests were considered significant if the *p*-value was less than 0.05 (* *p* < 0.05, *** *p* < 0.001). *p*-values greater than 0.05 are indicated as not significant (ns).

**Figure 4 ijms-24-13156-f004:**
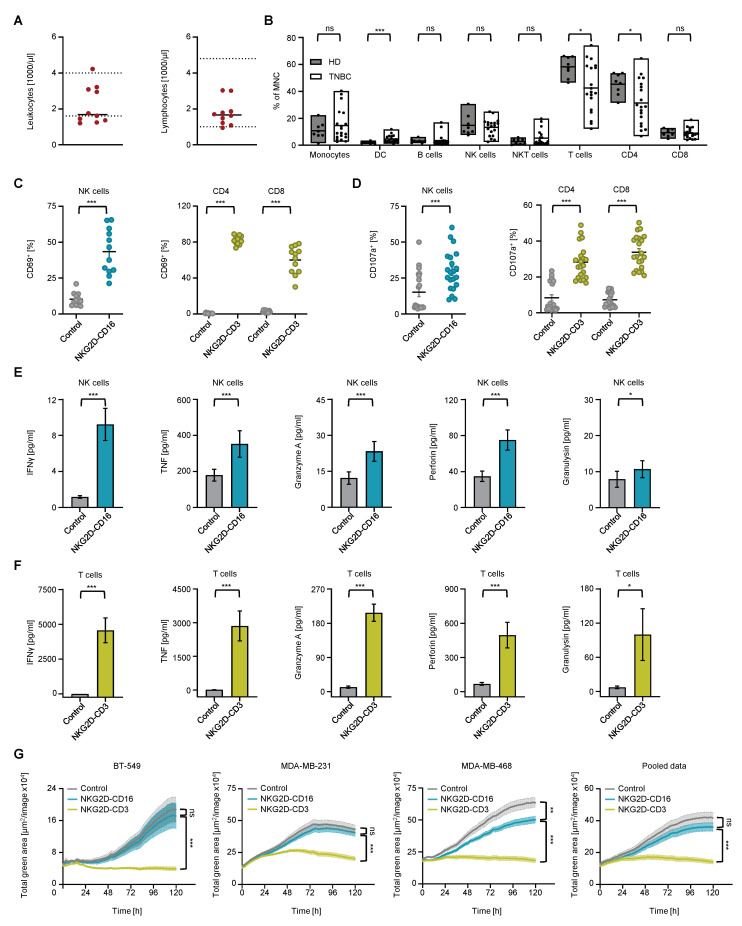
**Characterization of PBMCs from TNBC patients receiving chemotherapy or combined chemo–immunotherapy.** (**A**) Leukocyte and lymphocyte counts from TNBC patients (n = 10) at the time of sample collection are shown. Dotted lines indicate the normal range for healthy individuals. (**B**) Peripheral mononuclear cells (MNCs) from TNBC patients (n = 19) and healthy donors (n = 7) were identified by counterstaining for T cells (CD3^+^CD4^+^ and CD3^+^CD8^+^), monocytes (CD14^+^), dendritic cells (DCs, CD3^−^CD56^−^CD14^−^CD19^−^HLA–DR^+^), B cells (CD19^+^), NKT (CD3^+^CD56^+^) and NK cells (CD3^−^CD56^+^), and then analyzed by flow cytometry and presented as percentage of MNCs. (**C**–**F**) BT–549, MDA–MB–231 and MDA–MB–468 cells were cocultured with PBMCs from TNBC donors (n = 4) and controls (gray) or NKG2D–CD16 (blue) or NKG2D–CD3 (green) (both 2.5 µg/mL) at an E:T ratio of 2.5:1. (**C**) Activation of NK and CD4^+^ and CD8^+^ T cells as determined by expression levels of CD69 after 24 h and (**D**) degranulation of NK and CD4^+^ and CD8^+^ T cells as determined by expression levels of CD107a after 4 h were measured by flow cytometry. (**E**,**F**) Legendplex assays of supernatants were performed to analyze IFNγ, TNF, granzyme A, perforin and granulysin release at 24 h. (**G**) TNBC cell lysis by PBMCs from TNBC patients (n = 8) at an E:T ratio of 5:1 after control (gray) or NKG2D–CD16 (blue) and NKG2D–CD3 (green) treatment was measured using a live cell imaging system for 120 h. Data for indicated cell lines and combined results for 3 TNBC cell lines are shown. All statistical tests were considered significant if the *p*-value was less than 0.05 (* *p* < 0.05, ** *p* < 0.01, *** *p* < 0.001). *p*-values greater than 0.05 are indicated as not significant (ns).

**Table 1 ijms-24-13156-t001:** Clinical characteristics of TNBC patients and therapy line at the time of blood sampling.

Clinical Characteristics	Total (n = 19)	
	n (%)	Mean (Stdv.)
Gender Female Age, Mean Years	19 (100)	59.4 (±10.4)
**TNM classification**		
T1c N0 MX T1c N0 M0 T1c N1 MX T1c N1 M0 T2 N0 MX T2 N0 M0 T2 N1 MX T2 N1 M0 T2 N1 M1 T3 N0 MX T3 N1 M0 T3 N2a M0 T4d N1 MX	2 (11) 2 (11) 1 (5) 1 (5) 2 (11) 4 (21) 1 (5) 1 (5) 1 (5) 1 (5) 1 (5) 1 (5) 1 (5)	
**Histological subtype**		
*ER/PR status* negative <10% 10–50%	15 (79) 3 (16) 1 (5)	
*Her2 neu status* negative + ++	12 (63) 3 (16) 2 (11)	
**Histological grading**		
G2 G3	8 (42) 11 (58)	
**Therapy line at the time of blood sampling**		
Atezolizumab Carboplatin, Paclitaxel Carboplatin, Paclitaxel, Atezolizumab Carboplatin, Paclitaxel, Pembrolizumab Cyclophosphamide, Epirubicin Denosumab Denosumab, Paclitaxel, Pembrolizumab Sacituzumab	1 (5) 3 (16) 1 (5) 7 (37) 2 (11) 1 (5) 1 (5) 1 (5)	
**Days since last treatment dose**		
d7 d8 d14 d21 d34	12 (63) 2 (11) 1 (5) 2 (11) 1 (5)	
**Peripheral blood count**		
Leukocytes (1000/µL) Lymphocytes (1000/µL) Monocytes (1000/µL) Neutrophils (1000/µL) Platelet count (1000/µL) Hb (g/dL)		5.2 (±2.4) 1.8 (±0.6) 0.5 (±0.15) 3.1 (±1.9) 275 (±143) 11.1 (±0.9)

## Data Availability

The datasets supporting the conclusions of this article are included within the article.

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
