# Peer review of "Targeting NKG2DL with Bispecific NKG2D–CD16 and NKG2D–CD3 Fusion Proteins on Triple–Negative Breast Cancer"

_ijms, 2023, doi:10.3390/ijms241713156_

Round 1

Reviewer 1 Report

This article reports the use of bispecific fusion proteins (BFPs) consisting of the NKG2D receptor domain fused with either anti-CD3 (NKG2D-CD3) or anti-CD16 (NKG2D-CD16) Fab fragment to target triple-negative breast cancer (TNBC). The novelty of this study was to apply the BFPs effectively direct NK cells and T cells toward the tumor cells and eliminate the TNBC cells. However, there are several minor concerns as follows:

1.    There are so many typos. The manuscript is suggested for English language corrections and improvements.

2.    Please reformat the manuscript based on the guidance of IJMS.

3.    Materials and Methods: In the section Production and purification of bispecific NKG2D fusion proteins, please give the protocol in detail and show the SDS-PAGE and Endotoxin test in the Supp.

4.    Please cross-check the references in the list of references and citations in the text.

Extensive editing of English language required

Author Response

Point to Point Reply

Reviewer#1

This article reports the use of bispecific fusion proteins (BFPs) consisting of the NKG2D receptor domain fused with either anti-CD3 (NKG2D-CD3) or anti-CD16 (NKG2D-CD16) Fab fragment to target triple-negative breast cancer (TNBC). The novelty of this study was to apply the BFPs effectively direct NK cells and T cells toward the tumor cells and eliminate the TNBC cells. However, there are several minor concerns as follows:

  1. There are so many typos. The manuscript is suggested for English language corrections and improvements.

We apologize for any typographical errors that may have been present in the original version. We have carefully revised the manuscript, addressing all linguistic issues, including the correction of typographical errors. These revisions are highlighted in yellow.

  1. Please reformat the manuscript based on the guidance of IJMS.

Please find attached the reformatted version of the manuscript.

  1. Materials and Methods: In the section Production and purification of bispecific NKG2D fusion proteins, please give the protocol in detail and show the SDS-PAGE and Endotoxin test in the Supp.

We thank the reviewer for this request. We apologize that we didn't include this information in our original version, as it provides a better understanding of our experimental procedures. You can now see the SDS gel and endotoxin assay figures as Supplementary Figure 5 A and B, respectively.

  1. Please cross-check the references in the list of references and citations in the text.

We thank the reviewer for bringing this matter to our attention, and we are pleased to confirm that the references have been carefully reviewed and adjusted in accordance with your feedback.

Reviewer 2 Report

Kaidun et al created bispecific fusion proteins with the NKG2D receptor domain linked to anti-CD3 or anti-CD16 Fab-fragments and studied the effects of these fusion proteins on directing natural killer cells and T cells to Triple negative breast cancer (TNBC) cells and eliminating TNBC cells. The first characterized NKG2DL Expression in selected triple negative breast cancer cells. Then they checked whether the fusion proteins could recruit natural killer cells and T cells from healthy donors to TNBC cells and lyse these TNBC cells. Finally, they demonstrated that PBMC from TNBC patients that received chemotherapy could also be recruited to TNBC cells and efficiently remove the TNBC cells. This is a nice manuscript, and the results indicate that the fusion proteins may be used alone or together with chemotherapy for treating TNBC tumors. The manuscript is good for publication after resolving the minor issues below.

Minor comments:

1.     Page 2, paragraph 5, line 5: “BT-549 three types of NKG2DL (each 12.5%)” should be “BT-549 expressing three types of NKG2DL (each 12.5%)”

2.     Figure 3I, Figure 3J: Please label each panel with the TNBC cell lines like figure 3H.

3.     Figure S3B: label each panel with the TNBC cell lines like figure S3A.

4.     Figure 2 legend: It was mentioned that “PBMC of healthy donors in coculture with or without TNBC cells at an E:T ratio of 2.5:1 in the presence or absence of NKG2D-CD16/CD3 (2.5 μg/ml)”. However, no data for activation of NK cells and T cells without coculture with TNBC cells was shown. Did TNBC cells affect the activation of NK cells and T cells?

5.     Like comment 4, figure 3 legend, figure S3 legend and figure S4 legend also mentioned the “coculture of PBMC with or without TNBC cells to detect reactivity of T and NK cells against TNBC cells. Again, no data was shown for coculture without TNBC cells. Are there typos in these descriptions? Because you need TNBC cells to check the reactivity of T and NK cells against TNBC cells.

6.     Method section: “Production and purification of bispecific NKG2D fusion proteins”: The method description is highlighted in bold. Please remove the bold format.

Author Response

Point to Point Reply

Reviewer#2

Kaidun et al created bispecific fusion proteins with the NKG2D receptor domain linked to anti-CD3 or anti-CD16 Fab-fragments and studied the effects of these fusion proteins on directing natural killer cells and T cells to Triple negative breast cancer (TNBC) cells and eliminating TNBC cells. The first characterized NKG2DL Expression in selected triple negative breast cancer cells. Then they checked whether the fusion proteins could recruit natural killer cells and T cells from healthy donors to TNBC cells and lyse these TNBC cells. Finally, they demonstrated that PBMC from TNBC patients that received chemotherapy could also be recruited to TNBC cells and efficiently remove the TNBC cells. This is a nice manuscript, and the results indicate that the fusion proteins may be used alone or together with chemotherapy for treating TNBC tumors. The manuscript is good for publication after resolving the minor issues below.

We thank the reviewer for this positive evaluation of our manuscript. We have carefully revised the manuscript according to the issues raised below and highlighted the changes in yellow.

  1. Page 2, paragraph 5, line 5: “BT-549 three types of NKG2DL (each 12.5%)” should be “BT-549 expressingthree types of NKG2DL (each 12.5%)”

The missing word “expressing” was added in the revised version of the manuscript.

  1. Figure 3I, Figure 3J: Please label each panel with the TNBC cell lines like figure 3H.

We thank the reviewer for raising this point. In response to your suggestion, we have diligently labeled each panel of Figure 3I and Figure 3J with the appropriate TNBC cell line designations.

  1. Figure S3B: label each panel with the TNBC cell lines like figure S3A.

Like in the previous point we added the proper labeling to Figure S3B.

  1. Figure 2 legend: It was mentioned that “PBMC of healthy donors in coculture with or without TNBC cells at an E:T ratio of 2.5:1 in the presence or absence of NKG2D-CD16/CD3 (2.5 μg/ml)”. However, no data for activation of NK cells and T cells without coculture with TNBC cells was shown. Did TNBC cells affect the activation of NK cells and T cells?

We thank the reviewer for this important point and apologize for any misunderstanding that may have caused by the inaccuracy of the legend. Given the nature of our study and the intended scope of our findings, we did not aim to assess immune cell activation in the absence of TNBC cells. All necessary corrections according to your comments were highlighted in yellow. 

  1. Like comment 4, figure 3 legend, figure S3 legend and figure S4 legend also mentioned the “coculture of PBMC with or without TNBC cells to detect reactivity of T and NK cells against TNBC cells. Again, no data was shown for coculture without TNBC cells. Are there typos in these descriptions? Because you need TNBC cells to check the reactivity of T and NK cells against TNBC cells.

We amended the manuscript as stated in response to point 4. of the reviewer.

  1. Method section: “Production and purification of bispecific NKG2D fusion proteins”: The method description is highlighted in bold. Please remove the bold format.

The bold format was removed.